# Improvement of the Enterotoxigenic *Escherichia coli* Infection Model for Post-Weaning Diarrhea by Controlling for Bacterial Adhesion, Pig Breed and *MUC4* Genotype

**DOI:** 10.3390/vetsci7030106

**Published:** 2020-08-07

**Authors:** Hiroki Matsumoto, Masashi Miyagawa, Sayaka Takahashi, Ryouichi Shima, Takayuki Oosumi

**Affiliations:** 1Research and Development Section, Institute of Animal Health, JA Zen-Noh (National Federation of Agricultural Cooperative Associations), 7 Ohja-machi, Sakura-shi, Chiba 285-0043, Japan; miyagawa-masashi@zennoh.or.jp (M.M.); oosumi-takayuki@zennoh.or.jp (T.O.); 2Diagnostic Center, Institute of Animal Health, JA Zen-Noh (National Federation of Agricultural Cooperative Associations), 7 Ohja-machi, Sakura-shi, Chiba 285-0043, Japan; takahashi-sayaka@zennoh.or.jp; 3JA Zen-Noh (National Federation of Agricultural Cooperative Associations), 1-3-1 Ootemachi, Chiyoda-ku, Tokyo 100-6832, Japan; shima-ryouichi@zennoh.or.jp

**Keywords:** enterotoxigenic *E. coli*, post-weaning diarrhea model, pig, *MUC4* genotype, experimental infection

## Abstract

Enterotoxigenic *Escherichia coli* (ETEC) is a major cause of post-weaning diarrhea (PWD) in pigs and causes significant damage to the swine industry worldwide. In recent years, there has been increased regulation against the use of antibacterial agents in swine due to their health risks. Utilizing experimental models that consistently recapitulate PWD is important for the development of non-antibacterial agents against PWD in pigs. In this study, we established a highly reproducible PWD infection model by examining differences in adhesion of ETEC to the intestinal tissue as well as the association between *MUC4* polymorphisms and sensitivity to PWD. Post-weaning diarrhea differences between pig breeds were also examined. The adhesion to enterocytes varied from 10^4.0^ to 10^6.4^ CFU/mL even among the F4 ETEC strains. Experimental infection revealed that PWD can be induced in all *MUC4* genotypes after infection with 10^10^ CFU/pig of highly adherent ETEC, although there were variable sensitivities between the genotypes. Lowly adherent ETEC did not cause PWD as efficiently as did highly adherent ETEC. The incidence of PWD was confirmed for all pigs with the ETEC-susceptible *MUC4* genotypes in all of the breeds. These results indicate that high-precision and reproducible experimental infection is possible regardless of pig breeds by controlling factors on the pig-end (*MUC4* genotype) and the bacterial-end (adhesion ability).

## 1. Introduction

Post-weaning diarrhea (PWD) is most often caused by enterotoxigenic *Escherichia coli* (ETEC) in piglets 3–10 days after weaning. Pigs infected with ETEC typically exhibit watery diarrhea lasting one to five days, causing death by dehydration or growth deterioration in the surviving piglets and resulting in serious economic loss in the swine industry worldwide [1,2]. Enterotoxigenic *E. coli* that induces PWD produces heat-labile enterotoxin (LT) and/or heat-stable enterotoxin (STa and STb) and have fimbriae required to colonize intestinal cells. Enterotoxigenic *E. coli* exhibits mainly F4 and F18 fimbriae in pigs. Although F5, F6 and F41 fimbriae have been reported, their separation frequency is low [3]. While there are multiple known variants of F4 (F4ab, F4ac and F4ad) and F18 (F18ab and F18ac) fimbriae, the main variants seen in PWD are F4ac and F18ac [4,5]. The gene mucin 4 (*MUC4*) encodes for the F4 fimbriae receptor. The *MUC4* gene has been identified on chromosome 13 [6,7,8]. Recently, a DNA marker-based test targeting *MUC4* has become available to genotype pigs and identified single nucleotide polymorphism (SNP) (DQ848681: g.8227C > G) associated with F4ab/ac ETEC resistance and susceptibility [9]. Using this DNA marker-based test to genotype for *MUC4*, pigs can be categorized into the following three groups in terms of their susceptibility to ETEC-induced diarrhea: resistant (RR), susceptible heterozygote (SR) and susceptible homozygote (SS). While the incidence of diarrhea was increased by ETEC infection, sensitivity to PWD was dependent on *MUC4* genotype and the occurrence of diarrhea did not reach 100% [10,11]. As differences in *MUC4* genotypes appear to affect the composition of gastrointestinal microbiota, *MUC4* polymorphisms have been proposed to play roles in intestinal infection [12], such as PWD.

Antimicrobial agents have been used to control PWD. However, the emergence of antimicrobial resistance bacteria has been regarded as a global problem and has necessitated alternatives to antimicrobial agents against PWD [13,14,15]. Therefore, establishing a reproducible and highly efficient infection model for ETEC diarrhea using readily available pigs is important for the assessment of non-antimicrobial agents against PWD.

In this study, the ability of ETEC to adhere to enterocytes was first evaluated using various ETEC strains established in Japan. Furthermore, lowly or highly adherent ETEC strains were inoculated into pigs with different *MUC4* polymorphisms, as well as into different breeds of the same *MUC4* genotype. We aimed to further improve the PWD infection model by examining the frequency and degree of the resulting PWD.

## 2. Materials and Methods

### 2.1. Bacterial Strains

One hundred ETEC strains were isolated from pigs in the JA Zen-noh Diagnostic Center (Sakura, Japan). Briefly, feces from pigs were plated on 5% sheep blood agar followed by overnight cultivation, and coliform colonies with hemolytic activity were isolated. Biochemical properties (Triple Sugar Iron (TSI) agar medium, sulfide indol motility (SIM) agar medium, indole-producing ability) were further characterized for the confirmation of *E. coli*. The ETEC strains were stored at −80 °C in 10% skim milk. The strains were recultured on deoxycholate hydrogen sulfide lactose (DHL) agar at 37 °C overnight before each experiment.

### 2.2. Preparation of DNA Template and PCR Amplification

From each strain, bacterial DNA was extracted using InstaGene Matrix (Bio-Rad Laboratories, Tokyo, Japan), followed by a check for DNA quality and purity using a NanoDrop (Thermo Scientific, Tokyo, Japan). The extracted DNA was kept at −20 °C until used for PCR. The PCR mixture consisted of 2 µL of the DNA template, 1-µM primers (Table 1), 1.25 U Ex Taq HS (TaKaRa Bio, Inc. Kusatsu, Japan), 5 µL of 10× PCR Buffer (Mg^2+^ plus), 200 µM of dNTP mixture and up to 50 µL of distilled water. After activation at 94 °C for 2 min, the reactions were subjected to 30 cycles of amplification consisting of a 30 s denaturation at 94 °C, 30 s annealing at 55 °C, 60 s extension at 72 °C and a final extension step at 72 °C for 10 min. For detection of the PCR products, amplified DNA samples were examined by electrophoresis in a 2% agarose gel and visualized with ethidium bromide under UV light.

### 2.3. Animals

All animal studies were performed at the JA Zen-noh Institute of Animal Health after being approved by the Animal Experimental Review Committee (ethical approval code No. 324 and 366). WL (large white × Landrace), LW (Landrace × large white), LL (Landrace × Landrace), WW (large white × large white) and LWD (Landrace × large white × Duroc) pigs were originated from specific pathogen-free herds. All pigs were transported into the Zen-noh Institute of Animal Health at day of weaning (21 day of age) and housed in individual pens. Body weights were measured at 21 day of age, autopsy or death. Pigs were fed a standard diet of approximately 400 g per day and were provided water ab libitum via push drinker.

### 2.4. Porcine Intestinal Epitheliocyte (PIE) Cell Line

Porcine intestinal epitheliocyte cells are non-transformed intestinal cultured cells originally derived from intestinal epithelia isolated from an unsuckled neonatal swine [17]. Porcine intestinal epitheliocyte cells were maintained in antibiotic free Dulbecco’s modified Eagle’s medium (DMEM, High Glucose, Pyruvate) (Gibco, Tokyo, Japan) supplemented with 10% fetal bovine serum (FBS) at 37 °C in 5% CO_2_.

### 2.5. Bacterial Adherence Assays

Porcine intestinal epitheliocyte cells were cultured in 6-well plates (approximately 5 × 10^5^ cells/well) to confluence. Each ETEC strain was grown in Luria–Bertani (LB) medium for 6 h at 37 °C and harvested. After centrifugation, the cells were washed three times in sterile phosphate buffered saline (PBS). Adherence assays were performed by inoculating ~1 × 10^7^ CFU/mL of each ETEC on a confluent PIE cell monolayer. The plates were incubated for 1 h at 37 °C in 5% CO_2_. The monolayers were then washed with PBS, trypsinized and disrupted by repeated pipetting. Serial dilutions of the cell lysates were plated onto deoxycholate hydrogen sulfide lactose (DHL) agar plates and incubated overnight at 37 °C for bacterial enumeration.

### 2.6. DNA Marker-Based Test

Piglets were tested for their sensitivity against the F4ab/ac fimbriae of *E. coli* by a genetic marker test for *MUC4* [9]. Tails samples were taken from the piglets shortly after farrowing, and genomic DNA was extracted using the QIAamp DNA Minikit (QIAGEN, Tokyo, Japan) according to the manufacturer’s instructions. The PCR mixture consisted of 1 µL of DNA template, 0.1-µM primers (Table 1), 0.5 U Ex Taq HS (TaKaRa Bio, Inc., Shiga, Japancity), 2.5 µL of 10× PCR Buffer (Mg^2+^ plus), 200 µM of dNTP mixture and water up to 25 µL. After activation at 95 °C for 15 min, the reactions were subjected to 35 cycles of amplification consisting of a 15-s denaturation at 95 °C, 30 s annealing at 60 °C, 30 s extension at 72 °C and a final extension step at 72 °C for 10 min. Touchdown PCR was performed, whereby the annealing temperature was decreased by 1 °C per cycle in the first 10 cycles, starting at 60 °C. The last 25 cycles were performed at an annealing temperature of 50 °C. The amplified DNA was digested with *XbaI* (TaKaRa Bio, Inc., Shiga, Japancity) at 37 °C for 2 h for restriction fragment length polymorphism (RFLP) analysis. The samples for RFLP analysis were examined by electrophoresis in a 2% agarose gel and visualized with ethidium bromide under UV light. The resistant allele (R) is indigestible by *XbaI* (367 base pairs), whereas the susceptible allele (S) is digested into two fragments (151 and 216 base pairs). The resulting genotypes will be henceforth referred to as SS (151 and 216 base pairs), SR (151, 216 and 367 base pairs) or RR (367 base pairs).

### 2.7. Bacterial Culture and Infection

From the results of the cell adhesion test, highly and lowly adherent (HA, LA) ETEC strains were selected and used as challenge strains. Artificial rifampicin resistance of these strains was induced via successive cultivation in DHL agar containing up to 100 µg/mL of rifampicin (RifDHL). These ETEC strains were grown in LB medium for 6 h at 37 °C, harvested by centrifugation and were resuspended with 10% skim milk. A total of 28 male pigs (WL) were assigned into 6 experimental groups based on their *MUC4* genotypes. They were orally challenged with the indicated amount of ETEC with a single dose at 22 days of age (Group 1: HA-ETEC, 10^10^ CFU per pig, SS genotype, *n* = 4; Group 2: HA-ETEC, 10^10^ CFU per pig, RS, *n* = 8; Group 3: HA-ETEC, 10^10^ CFU per pig, RR, *n* = 4; Group 4: HA-ETEC, 10^8^ CFU per pig, RS, *n* = 4; Group 5: LA-ETEC, 10^10^ CFU per pig, SS, *n* = 4; Group 6: LA-ETEC, 10^10^ CFU per pig, RS, *n* = 4). In another set of experiments, only the HA-ETEC strain was utilized in male RS pigs (WL, *n* = 5; LW, *n* = 5; LL, *n* = 12; WW, *n* = 10; LWD, *n* = 5). They were orally challenged by ETEC with a single dose of approximately 10^10^ CFU per pig at 22 days of age. At 7 days after infection, pigs were euthanized using sodium pentobarbital under sedation with xylazine and midazolam for both infection experiments.

### 2.8. Diarrhea Evaluation

The feces of the pigs were observed individually every morning. Fecal properties were measured as follows: 0 (normal), 1 (soft feces), 2 (semi-liquid) and 3 (liquid). The diarrhea incidence indicates the percentage of individuals exhibiting a score of 1 or more in each group. The duration of diarrhea was recorded individually, and the mean duration for the group was calculated. Mortality rate throughout the monitoring period was recorded.

### 2.9. Bacterial Shedding

Samples were collected from pigs using rectal swabs and were suspended in 1 mL sterile physiological saline at 1, 3, 5, 7 days post infection (dpi). The samples were serially diluted in sterile physiological saline, plated on RifDHL and incubated for 24 h at 37 °C. Colonies that grew on agar plates were directly counted.

### 2.10. Statistical Analyses

The mean fecal score was analyzed with repeated-measures ANOVA, with days post infection as a within subject factor and test groups as the between-subject, adjusting for pairwise comparisons by Bonferroni test. In terms of the incidence of diarrhea, pigs with a fecal score of 1 or more during the experimental period were considered to have diarrhea. The incidence of diarrhea and mortality were analyzed using Fisher’s exact test for each test group. Statistical analyses were performed using Excel 2016 (Microsoft Corporation, Washington, WA, USA) with the add-in software Statcel 4. A *p*-value of less than 0.05 was considered to be significantly different between groups. Results are presented as means ± SE, except the mortality rate.

## 3. Results

### 3.1. Detection of Fimbriae and Enterotoxins Genes

Of the 100 strains used in this study, 83 strains expressed the F4 fimbriae gene, 13 strains expressed the F18 fimbriae gene, and 4 strains (F_UT) did not express any genes corresponding to F4/F5/F6/F18/F41 fimbriae (Table 2). Furthermore, the strains carrying the F4 fimbriae gene also had almost all of the LT, STa and STb genes.

### 3.2. Bacterial Cell Adhesion Test

Among the strains having the F4 fimbriae gene, many were confirmed to be highly adherent (10^6^ CFU/mL or more) (Table 3). On the other hand, some F4 fimbriae strains had low adherence capacity of less than 10^5^ CFU/mL. It was observed that the F18 fimbriae and the F_UT strains exhibited low adherence capacities of 10^5^ CFU/mL or less. For the infection tests, highly adherent bacteria (adherence of 10^6.1^ CFU/mL, F4+, LT+, STa+, STb+ strains) and lowly adherent bacteria (adherence of 10^4.0^ CFU/mL, F4+, LT+, STa+, STb+) were individually selected.

### 3.3. Susceptibility to PWD in Pigs with Different MUC4 Genotypes Inoculated with Differentially Adherent F4 ETEC Strains

WL Pigs with different *MUC4* genotypes were inoculated with a single oral dose of ETEC 2 days after weaning. The mean fecal score was high at 1–3 dpi, and diarrhea was observed. Regardless of the degree of adhesion abilities of the strains, at 4–7 dpi, the surviving pigs tended to recover (Table 4). SS-genotyped pigs showed the most severe diarrheal symptoms (mean fecal score 2.00) in the group infected with highly adherent bacteria and challenged with 10^10^ CFU/pig, and the incidence of diarrhea was 100%. RS-genotyped pigs showed no significant differences in fecal score (1.83), incidence of diarrhea (100%) and diarrhea duration (3.4 days) compared to SS-genotyped pigs. On the other hand, the mean fecal score of the RR-genotyped pigs at 1–3 dpi was 0.25, significantly lower than that of SS- and RS-genotyped pigs, and the incidence of diarrhea was only 25%. In the group infected with highly adherent bacteria and challenged with 10^8^ CFU/pig, even RS-genotyped pigs had a low fecal score of 0.50, and the incidence of diarrhea was only 50%. In the group infected with lowly adherent bacteria and challenged with 10^10^ CFU/pig, SS- and RS-genotyped pigs had low fecal scores of 1.22 and 0.50, respectively, and the diarrheal incidence rate was only 50%.

The number of bacteria shedding during the test period is shown in Figure 1. In SS pigs infected with highly adherent bacteria and challenged with 10^10^ CFU/pig, the average bacterial shedding count of 10^7^ CFU/rectal swab and the high shedding continued to 5 dpi. RS pigs infected with highly and lowly adherent bacteria and challenged with 10^10^ CFU/pig showed 10^6^ to 10^7^ CFU/rectal swab and bacterial shedding up to 5 dpi. In all test groups, bacteria were shed at 10^4^ CFU/rectal swab or less by 7 dpi.

### 3.4. Differences in Susceptibility to PWD in Various RS-Type Pig Breeds Inoculated with Highly Adherent F4 ETEC

We next evaluated the differences in PWD sensitivity among different pig breeds with the RS genotype. A single oral administration of the HA (highly adherent) strain to 22-day-old RS pigs (WL, LW, LL, WW and LWD breeds) after weaning caused diarrhea (with the mean fecal scores at 1–3 dpi: 2.76 for LL, 2.75 for LWD and 2.47 for WW) (Table 5). Mean fecal scores for WL and LW were 1.82 and 1.50, respectively and were significantly lower than those of LL and LWD. In addition, the mean fecal scores at 4–7 dpi indicated that diarrhea in LL was more likely to occur than in other breeds. LL and LWD had similar mean fecal scores, but differences in mortality were noted. LL mortality was 75% and LWD was only 20%.

Figure 2 shows the number of bacterial shedding during the test period. The LL breed, which exhibited the highest mean fecal score and mortality rate, showed bacterial shedding of 10^8^ CFU/rectal swab at 1–3 dpi. For the other breeds, no difference was found in the number of bacterial shedding.

## 4. Discussion

In this study, we characterized the properties of 100 ETEC strains according to their fimbriae (F4, F5, F6, F18, F41) and enterotoxin (LT, STa, STb) gene expression. Among the strains examined, the strain carrying the F4 fimbrial gene was the most common (83/100; 83%) in the examined strains. According to other reports, many *E. coli* strains isolated from pigs express F4 as an adhesion gene, which is consistent with the present result [18,19,20]. However, Matsuda et al. and Jin et al. previously reported the rate of presence of the F4 fimbriae gene as 45.8% and 51.2%, respectively [19,20]. These differences may be due to the fact that our study targeted strains without the *stx2e* gene. On the other hand, strains carrying both the F4 and F18 genes were reported in a previous study [21]. Similar to previous studies [18,19,20], *E. coli* harboring the F4 gene was shown to be one of the major causes of PWD in this study. Furthermore, *E. coli* carrying all of the enterotoxin genes (LT, STa, STb) regardless of the fimbriae genes was mainly detected. It is still unclear the contribution of each individual enterotoxin strain to PWD; however, it is believed that all strains can cause diarrhea [22,23,24]. Therefore, we examined cell adhesion ability in strains carrying all toxin genes.

We found that F4 ETEC strains in general exhibit high adhesion ability (10^6^ CFU/mL) (Table 3). However, some of these strains also have a low adhesion ability (10^4.0^ to 10^4.9^ CFU/mL). This suggests that even ETECs with the same F4 gene had different adhering ability depending on the strain. In F18 ETEC strains, none exhibited high attachment ability. One possibility is that the F18 receptor may not be sufficiently expressed during the neonatal period, as PIE cells used in this test were established from newborn pigs [3,25,26]. Therefore, to evaluate the ability of F18 ETEC to adhere to cells, it will be important to utilize other experimental paradigms in future studies. In this study, we selected highly and lowly adherent strains in F4 ETEC and tested whether in vitro strain selection affects the experimental infection test.

In SS type pigs infected with 1010 CFU/pig of highly adherent ETEC, severe diarrhea (watery stool) was observed at 1–3 dpi, but not at 4–7 dpi (Table 4). However, the onset of diarrhea (score 1 or more) was 4.8 days, the same as previous reports [27]. In this study, we presumed that changes in fecal properties appear significantly at 1–3 dpi, and we focused on fecal scores for this 1–3 dpi period thereafter. Similar to previous reports [10,11], in the group infected with highly adherent bacteria and 10^10^ CFU/pig, it was shown that the sensitivity to F4 *E. coli* differs depending on the *MUC4* genotype: 2.00 for SS pigs, 1.83 for RS pigs and 0.25 for RR pigs. In the group infected with highly adherent bacteria and 10^8^ CFU/pig, the average fecal score of RS pigs was as low as 0.50, indicating that the number of bacteria was also an important factor. On the other hand, in the group infected with low adherence bacteria and 10^10^ CFU/pig, the incidence of diarrhea was clearly reduced, compared with the group infected with highly adherent bacteria and 10^10^ CFU/pig: 1.22 in SS pigs and 0.50 in RS pigs. These results indicate that it is very important to consider the adherence in selecting a challenging strain. Experimental infection studies suggested that a pig infection model for PWD could be established by infecting 10^10^ CFU/pig with highly adherent ETEC using SS or RS pigs (Table 4).

We evaluated the differences in PWD sensitivity among pig breeds with the RS genotype (Table 5 and Figure 2). WL pigs with the RS genotype were used in two independent infection experiments (Table 4 and Table 5), and their mean fecal scores at 1–3 dpi were 1.82 and 1.82, respectively, indicating high reproducibility. There were significant differences in the degree of diarrhea and marked differences in mortality among the breeds (Table 5). LL showed the highest diarrhea score (2.76), and a high mortality rate (75%). LWD and WW showed high fecal scores of 2.75 and 2.47, respectively. However, mortality rates of LWD and WW were lower (20% and 10%, respectively) than LL. These results suggest that the differences among breeds may influence their susceptibility to F4 ETEC even if they have the same *MUC4* genotype. It has been reported that the susceptibility to F4 ETEC is affected not only by the *MUC4* genotype, but also by other factors such as *MUC13* polymorphism, which may have contributed to differences in diarrhea and mortality among breeds [16,28]. Moreover, there have been reports that susceptibility to other diseases differ among breeds such the reproductive and respiratory syndrome virus, porcine circovirus type 2 [29,30]. However, in this study, we successfully induced diarrhea in all WL pigs carrying the RS and SS genes and in all pig breeds carrying the RS gene, albeit with varying degrees of diarrhea. Therefore, with this development of a highly accurate PWD infection model, we recommend the use of pigs with the *MUC4* gene (RS or SS) infected with 10^10^ CFU/pig with highly adherent ETEC.

## 5. Conclusions

It is clear that high-precision and reproducible experimental infection is possible regardless of pig breeds by controlling factors on the pig-end (*MUC4* genotype) and factors on the bacterial-end (adhesion ability). We believe that this study has further improved the accuracy of the PWD infection model. Our PWD infection model makes possible the development and evaluation of preventive agents against PWD without increasing the number of experimental animals.

## Figures and Tables

**Figure 1 vetsci-07-00106-f001:**
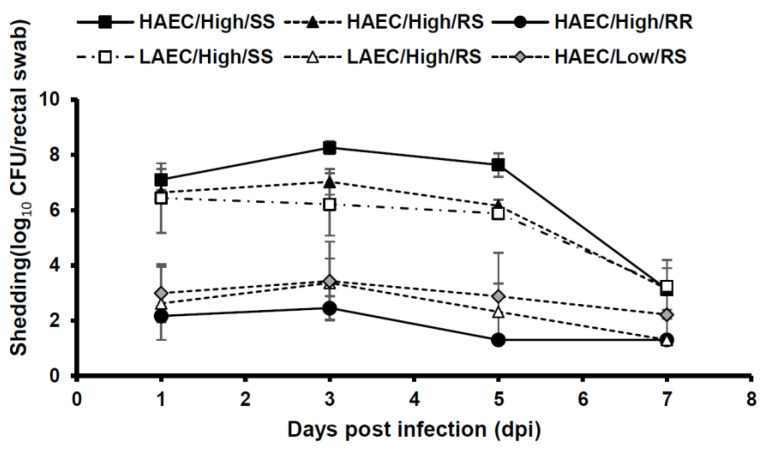
Mean F4 enterotoxigenic *Escherichia coli* (ETEC) shedding per rectal swab in experimental infections after oral challenge comparing highly and lowly adhering strains and *MUC4* genotypes. HEC and LEC indicate highly and lowly adherent *Escherichia coli*, respectively. High or Low indicates the amount of ETEC used to orally challenge the pigs (high—approximately 10^10^ CFU/pig; low—approximately 10^8^ CFU/pig). SS/RS/RR indicate the *MUC4* genotype of the pigs utilized. The detection limit was 200 CFU/rectal swab. Error bars, Standard error.

**Figure 2 vetsci-07-00106-f002:**
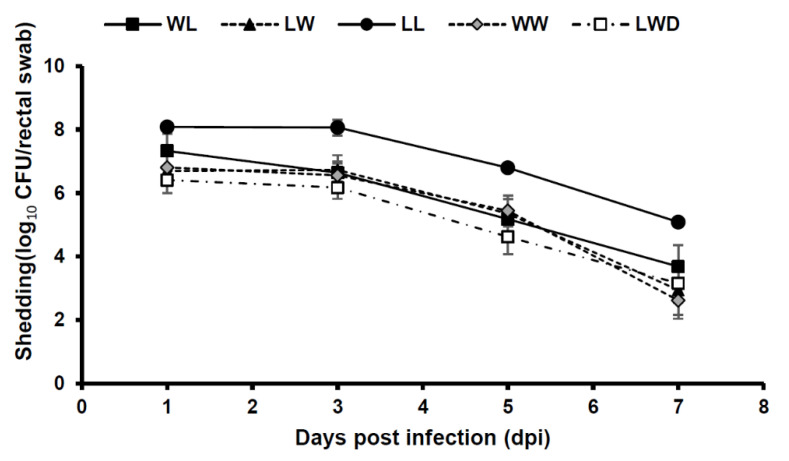
Mean F4 ETEC shedding per rectal swab in experimental infections after oral challenge comparing pig breeds with *MUC4* RS genotype. RS pigs [WL (large white × Landrace), LW (Landrace × large white), LL (Landrace × Landrace), WW (large white × large white) and LWD (Landrace × large white × Duroc)] were orally challenged with a single dose of ETEC (approximately 10^10^ CFU/pig). The amount of bacterial shedding was evaluated during the test period. Detection limit was 200 CFU/rectal swab. Error bars, Standard error.

**Table 1 vetsci-07-00106-t001:** Primers used in this study.

Target Gene	Primer Sequence (5’ to 3’)	Product Size (bp)	Reference
F4	F:GGTGATTTCAATGGTTCGGTC	704	[16]
	R:ATTGCTACGTTCAGCGGAGCG		
F5	F:TGCGACTACCAATGCTTCTG	450	[16]
	R:TATCCACCATTAGACGGAGC		
F6	F:TCTGCTCTTAAAGCTACTGG	333	[16]
	R:AACTCCACCGTTTGTATCAG		
F18	F:GTGAAAAGACTAGTGTTTATTTC	510	[17]
	R:CTTGTAAGTAACCGCGTAAGC		
F41	F:GAGGGACTTTCATCTTTTAG	431	[16]
	R:AGTCCATTCCATTTATAGGC		
LT	F:ATTTACGGCGTTACTATCCTC	280	[16]
	R:TTTTGGTCTCGGTCAGATATG		
STa	F:TCCGTGAAACAACATGACGG	244	[16]
	R:ATAACATCCAGCACAGGCAG		
STb	F:GCCTATGCATCTACACAATC	278	[16]
	R:TGAGAAATGGACAATGTCCG		
muc4	F:GTGCCTTGGGTGAGAGGTTA	367	[9]
	R:CACTCTGCCGTTCTCTTTCC		

**Table 2 vetsci-07-00106-t002:** Presence of adhesion factor and enterotoxin genes in *E. coli* from pigs in Japan.

Fimbriae Gene	Enterotoxin Genes			Number of Strains
	LT	Sta	STb	
F4	+	+	+	77
	+	−	+	6
F18	+	+	+	9
	+	+	−	3
	+	−	−	1
F_UT	+	+	+	3
	+	+	−	1

F_UT: Strains that do not express any fimbrial genes for F4, F5, F6, F18, F41.

**Table 3 vetsci-07-00106-t003:** Identification of adhesion ability to PIE cells.

Fimbriae Gene	Range of Bacterial Adherence (log_10_ CFU/mL)	Number of Strains
F4	6.0–6.4	17
	5.0–5.9	51
	4.0–4.9	9
F18	4.0–4.9	5
	3.0–3.9	4
F_UT	4.0–4.9	2
	3.0–3.9	1

**Table 4 vetsci-07-00106-t004:** Evaluation of diarrhea in experimental infections comparing high and low adhesion ETEC and MUC4 genotypes. Data are presented mean ± SE.

Challenged Strains	HAEC ^(1)^				LAEC ^(2)^	
Infectious Bacterial Load (CFU/pig)	10^10^			10^8^	10^10^	
Pig’s Genotyped of *MUC4*	SS	RS	RR	RS	SS	RS
*n*	4	8	4	4	4	4
Mean body weight (kg)						
Initial	6.93 ± 0.49	6.51 ± 0.51	5.88 ± 0.13	6.00 ± 0.64	5.93 ± 0.47	6.30 ± 0.48
Final ^(3)^	6.75 ± 0.43	6.88 ± 0.54	6.75 ± 0.32	7.13 ± 0.72	6.25 ± 0.78	7.00 ± 0.54
Average daily gain (g)	−21.9 ± 16.4	19.3 ± 54.8	109.4 ± 29.9	140.6 ± 12.9	−53.1 ± 188.8	87.5 ± 38.5
Mean fecal score						
1–3 dpi	2.00 ± 0.28 ^a,b,c^	1.83 ± 0.25 ^d,e,f^	0.25 ± 0.18 ^a,d^	0.50 ± 0.29 ^b,e^	1.22 ± 0.43	0.50 ± 0.29 ^c,f^
4–7 dpi	0.88 ± 0.27	0.43 ± 0.14	0	0.38 ± 0.20 ^a^	0.25 ± 0.25	0
Diarrhea incidence (%)						
1–3 dpi	100	100 ^g^	25 ^g^	50	50	50
4–7 dpi	75	71.4 ^h,i^	0 ^h^	25	50	0 ^i^
Duration of diarrhea (day)	4.8 ± 0.9	3.4 ± 0.7	0.5 ± 0.5	1.5 ± 01.2	2.0 ± 2.0	0.8 ± 0.5
Mortality (%)	0	12.5	0	0	50	0

^(1)^ Highly adherent *E. coli*. ^(^^2)^ Lowly adherent *E. coli*. ^(3)^ Measured at necropsy (7 dpi) or death. (^a–i^) Same superscripts indicate significant differences between two groups (*p* < 0.05).

**Table 5 vetsci-07-00106-t005:** Evaluation of diarrhea in experimental infections challenged with Highly adherent *E. coli* comparing breeds with the MUC4 RS genotype. Data are presented mean ± SE.

Pigs Breed ^(1)^	WL	LW	LL	WW	LWD
*n*	5	5	12	10	5
Mean body weight (kg)					
Initial	6.44 ± 0.22	6.34 ± 0.20	6.03 ± 0.25	6.44 ± 0.36	6.66 ± 0.22
Final ^(2)^	6.70 ± 0.41	6.70 ± 0.25	5.67 ± 0.31	6.30 ± 0.37	6.60 ± 0.24
Average daily gain (g)	−25.8 ± 94.7	45.0 ± 20.0	−150.0 ± 54.1	−20.2 ± 29.8	−19.5 ± 43.0
Mean fecal score					
1–3 dpi	1.82 ± 0.26 ^a^^,b^	1.50 ± 0.24 ^c^^,d^	2.76 ± 0.09 ^a^^,c^	2.47 ± 0.12 ^d^	2.75 ± 0.26 ^b^
4–7 dpi	0.25 ± 0.13 ^e1^	0.25 ± 0.12 ^f1^	1.58 ± 0.34 ^e^^,f^^,g^	0.71 ± 0.17 ^g^	1.00 ± 0.27^b^
Diarrhea incidence (%)					
1–3 dpi	100	100	100	100	100
4–7 dpi	33.3	40.0	100	80.0	100
Duration of diarrhea (day)	4.0 ± 1.0	3.8 ± 0.6	5.0 ± 0.6	4.3 ± 0.3	5.5 ± 1.0
Mortality (%)	40.0	0	75.0	10.0	20.0

^(1)^ WL:Large White × Landrace; LW:Landrace × Large White; LL:Landrace × Landrace; and WW:Large White × Large White; LWD:Landrace × Large White × Duroc. ^(2)^ Measured at necropsy (7 dpi) or deat.h. (^a–g^) Same superscripts indicate significant differences between two groups (*p* < 0.05).

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
