# Peer review of "Improvement of the Enterotoxigenic Escherichia coli Infection Model for Post-Weaning Diarrhea by Controlling for Bacterial Adhesion, Pig Breed and MUC4 Genotype"

_vetsci, 2020, doi:10.3390/vetsci7030106_

Round 1

Reviewer 1 Report

In this manuscript the authors want to establish a highly reproducible PWD infection model by examining differences in adhesion of ETEC to the intestinal tissue as well as the association between MUC4 polymorphisms and differential sensitivity to PWD. However, we cannot understand how to set up an ETEC infection model and where to improve it within the text, such as the lack of basic information such as the number and body weight of pigs, the test period, etc. There is no basis for the success of the PWD infection model. Normally, we should measure body temperature and blood routine test.

Author Response

In this manuscript the authors want to establish a highly reproducible PWD infection Number and weight of pigs displayed in text and table as the association between MUC4 polymorphisms and differential sensitivity to PWD. However, we cannot understand how to set up an ETEC infection model and where to improve it within the text, such as the lack of basic information such as the number and body weight of pigs, the test period, etc. There is no basis for the success of the PWD infection model. Normally, we should measure body temperature and blood routine test.

Response:

Further details of the infection method are now described in the Materials and Methods (Sections 2.3. Animals and 2.7. Bacterial culture and infection), as requested. The number of pigs utilized and their body weights are indicated in the text, as well as in Tables 4 and 5.

In previous reports, 100% diarrhea incidence had not been accomplished even when using pigs of the ETEC-sensitive RS genotype. In this study, however, we succeeded in developing diarrhea in 100% of the RS pigs regardless of breed by infecting the pigs with 1010 CFU/pig of highly adherent ETEC. To our knowledge, there are not many studies using body temperature and blood tests as an indicator of successful infection; therefore, we evaluated the success of the PWD infection model by the development of diarrhea.

Reviewer 2 Report

Major comments.

Although the manuscript has some merits, I have some concerns about the design of the experiment and statistical analyses

Many references from the authors using MUC13 genes instead of MUC4 genes. I would not believe the assessment of susceptibility and resistance to animals can be done by a single gene (MUC 4). Especially, here the authors only used one SNP in the MUC 4 gene.  At least, the SNP used should be widely studied and validated in a large population, otherwise, it does not really make sense.

Since the authors used different breeds, but it was not clear in the manuscript how many animals in each breed. The sample size is relatively small to test many comparisons.

Many statistical analyses were done without justification and explanation, I would like to recommend the pairwise group comparison is suitable for analyses as the authors have three different groups of genotypes as well as many different time points. The statistical models should consider these factors in one analysis, GLM models should be used. Also, if the authors want to compare the breed, then the interaction between breed and genotypes should be included in the model.

Lastly, the authors should improve their writing, I have done the check for results and discussion section as there are many errors here.

Minor

Line 21: set out to establish =   established

Line 22: It is hard to get the idea of why the authors included MUC4 genes and how many SNPs from the gene were used

Line 22: what did the authors mean differential sensitivity

Line 23: Remove at the same time, it is not scientific style

Line 23-24: The authors can re-write as: The adhesion to enterocytes varies from  … even among the F4 ETEC strains

Line 25-26: It is hard to understand the content, please re-write it as well.

Line 25-26: Is diarrhea in the text same as PWD

The abstract is very hard to understand, some sentences are not in scientific styles. It is not clear between methods and results in the abstract. The concluding sentence in the abstract is not well linked to the rest.

Line 34, 37, 38, etc : Please do not start the sentence with an abbreviation

Line 34: The authors need to re-define the abbreviation

Line 37: From the definition of authors in the abstract, ETEC should be singular nor plural

Line 41-44: Please re-write the sentence or break it into two sentences

Line 44: “this method” which methods did the authors want to mention

Line 47: Please explore how they can do it in one or two sentences

Line 48: Why did authors talk about gut microbiota

Line 49: I think antimicrobial agents are still using now, please remove historically

Line 50: a problem around the world  =  global problem

Line 61: Please add the city and country for the company

Line 69: did the authors check the quality of DNA, how did the author check the purity of DNAs

Line 80 What are the different the authors between WL and LW

How many pigs used in the study?

Line 84:why did authors choose 400 g/day

Line 100:  If it has MUC4 then stated it, it is not clear what means a genetic marker here.

Line 110: How many bands the authors obtained after enzyme digestion

Please provide details about time and temperature for digestion

What is about the pigs with three bands “367. 151 and 216”

Line 137: it is hard to understand how the authors did statistical analyses:

-          Could the authors explore how Tukey-Kramer method was used

-          Why the authors select Fisher’s exact test in this case, which is the test group (which one versus which one).

-          Why did the authors use the t-test and which type of t-test?

-          Did the authors consider the interaction between the strain and genotypes?

-           

Table 3; The authors indicate the rank, not the mean in the second column

Line 161: I did not convince the author's possibility to use one SNP to define the animals as Susceptible or resistant to disease. Moreover, the authors should provide more details about this marker in the introduction and method sections

Table 4” Please bring the SE

Did the authors have the PCR-RLFP picture?

Since the authors have very hard to adjust the changes in the experiments are true or not

Table 4: it is not clear differences between the

Table 5: How many pigs in each breed

Please add the footnote for the breed name

Line 206: What is the mean of Bars and SE

Line 213: replace “studies by Matsuda et al. and Jin et a report” Matsuda et al. and Jin et al previously
reported

Line 214: Why the stx2e gene can cause the difference

Line 218 219: please re-write

Line 230-231: The authors did not need to repeat the results

Table 5: There is some difference between the WL and LL in the mean fecal score, duration, and mortality? What did the authors could explain it?

Author Response

Major comments.

Although the manuscript has some merits, I have some concerns about the design of the experiment and statistical analyses

Many references from the authors using MUC13 genes instead of MUC4 genes. I would not believe the assessment of susceptibility and resistance to animals can be done by a single gene (MUC 4). Especially, here the authors only used one SNP in the MUC 4 gene.  At least, the SNP used should be widely studied and validated in a large population, otherwise, it does not really make sense.

Response:

We appreciate the Reviewer’s feedback. It is worthwhile to mention that the purpose of this study was to develop a successful PWD infection model, not to investigate which genes are predominantly involved in the onset of PWD. We succeeded in achieving 100% incidence of diarrhea in RS-type pigs regardless of breed by administering highly adherent E. coli at 1010 CFU/pig. Using a PWD infection model with high accuracy for the development of diarrhea, such as our method, we are better equipped at developing countermeasures against PWD.

Since the authors used different breeds, but it was not clear in the manuscript how many animals in each breed. The sample size is relatively small to test many comparisons.

Response:

Further details of the infection method are now described in the Materials and Methods (Sections 2.3. Animals and 2.7. Bacterial culture and infection). The number of pigs utilized and their body weights are indicated in the text, as well as in Tables 4 and 5. Although the sample sizes may be small and the number of animals uneven between groups, we were able to successfully and meaningfully develop a reliable infection model using various genotypes.

Many statistical analyses were done without justification and explanation, I would like to recommend the pairwise group comparison is suitable for analyses as the authors have three different groups of genotypes as well as many different time points. The statistical models should consider these factors in one analysis, GLM models should be used. Also, if the authors want to compare the breed, then the interaction between breed and genotypes should be included in the model.

Response:

We conducted additional statistical analyses using the methods recommended by the Reviewer, and the results are reflected in the text and tables in the manuscript. Significant differences between groups were unchanged between the original (first version of manuscript) and suggested statistical analyses (revised manuscript). A more detailed response to the Reviewer is included in the minor comments section.

Lastly, the authors should improve their writing, I have done the check for results and discussion section as there are many errors here.

Response:

The original manuscript was reviewed by a native English speaker/writer with a Ph.D., and the revised text was also reviewed by the same individual.

Minor [Line number (original manuscript)/Line number (revised manuscript)]

Line 21/Line20: set out to establish =   established

We have modified the text in the revised manuscript according to the Reviewer’s recommendation.

Line 22/Line 22: It is hard to get the idea of why the authors included MUC4 genes and how many SNPs from the gene were used

For the current study, we focused on one SNP (MUC4 g.8227C>G) in the MUC4 gene. The reason why we chose MUC4 (and this particular SNP) is because it has been found to be associated with PWD based on previous reports and can be easily detected by PCR-RFLP.

Line 22/Line 22: what did the authors mean differential sensitivity

We have modified the text in the revised manuscript, replacing “differential sensitivity” with “sensitivity.”

Line 23/Line 23: Remove at the same time, it is not scientific style

We have modified the text in the revised manuscript according to the Reviewer’s recommendation.

Line 23-24/Line 23-24: The authors can re-write as: The adhesion to enterocytes varies from  … even among the F4 ETEC strains

We have modified the text in the revised manuscript according to the Reviewer’s recommendation.

Line 25-26/Line 26-27: It is hard to understand the content, please re-write it as well.

We have re-written the text in the revised manuscript according to the Reviewer’s recommendation.

Line 25-26/Line 26-27: Is diarrhea in the text same as PWD

Post-weaning pigs are more likely to develop diarrhea due to changes in the environment such as modifications in feed, and it has been reported that PWD likely affects later pig development. In this study, we used diarrhea and PWD synonymously, since only weaned pigs were used.

The abstract is very hard to understand, some sentences are not in scientific styles. It is not clear between methods and results in the abstract. The concluding sentence in the abstract is not well linked to the rest.

We have modified abstract in the revised manuscript, according to the Reviewer’s recommendation.

Line 34, 37, 38, etc/Line 42, 43, 46, etc  : Please do not start the sentence with an abbreviation

We have modified the text in the revised manuscript according to the Reviewer’s recommendation.

Line 34/Line 42, 43: The authors need to re-define the abbreviation

We have redefined the abbreviations in the revised manuscript according to the Reviewer’s recommendation.

Line 37/Line 46: From the definition of authors in the abstract, ETEC should be singular nor plural

We have modified the text in the revised manuscript according to the Reviewer’s recommendation.

Line 41-44/Line 51-53: Please re-write the sentence or break it into two sentences

We have re-written the indicated sentences in the revised manuscript according to the Reviewer’s recommendation.

Line 44/Line 53: “this method” which methods did the authors want to mention

We have made clarifications in the revised manuscript according to the Reviewer’s recommendation.

Line 47/Line 55-60: Please explore how they can do it in one or two sentences

We have the following details in the revised manuscript according to the Reviewer’s recommendation:

Line 48/Line 55-60: Why did authors talk about gut microbiota

To provide more context, we have modified the revised manuscript as follows:

While the incidence of diarrhea was increased by ETEC infection, sensitivity to PWD was dependent on MUC4 genotype and the occurrence of diarrhea did not reach 100% [7,8]. As differences in MUC4 genotypes appear to affect the composition of gastrointestinal microbiota, MUC4 polymorphisms have been proposed to play roles in intestinal infection [9], such as PWD.

Line 49/Line63: I think antimicrobial agents are still using now, please remove historically

We have modified the text in the revised manuscript according to the Reviewer’s recommendation.

Line 50/Line64: a problem around the world  =  global problem

We have modified the text in the revised manuscript according to the Reviewer’s recommendation.

Line 61/Line 75-76: Please add the city and country for the company

We have modified the text in the revised manuscript according to the Reviewer’s recommendation.

Line 69/Line 83-84: did the authors check the quality of DNA, how did the author check the purity of DNAs

We have modified the text in the revised manuscript according to the Reviewer’s recommendation, clarifying that the quality and purity of the DNA were checked via NanoDrop.

Line 80/Line 93: What are the different the authors between WL and LW

How many pigs used in the study?

WL indicates a pig born from a female Large White (W) and a male Landrace (L). In contrast, LW indicates a pig born from a female Landrace (L) and a male Large White (W). The number of pigs used were described in the text and tables.

Line 84/Line 100:why did authors choose 400 g/day

This is the standard diet given to 21-day-old piglets.

Line 100Line 116-117:  If it has MUC4 then stated it, it is not clear what means a genetic marker here.

We have made clarifications in the revised manuscript to indicate that the genetic marker that was tested for was MUC4.

Line 110/Line 126-131: How many bands the authors obtained after enzyme digestion

Please provide details about time and temperature for digestion

What is about the pigs with three bands “367. 151 and 216”

We have made the following clarifications in the revised manuscript according to the Reviewer’s recommendation:

The amplified DNA was digested with XbaI (TAKARA) at 37℃ for 2 hours for restriction fragment length polymorphism (RFLP) analysis. RFLP samples were examined by electrophoresis in a 2% agarose gel and visualized with ethidium bromide under UV light. The resistant allele (R) is indigestible by XbaI (367 base pairs), whereas the susceptible allele (S) is digested into two fragments (151 and 216 base pairs). The resulting genotypes will be henceforth referred to as SS (151 and 216 base pairs), SR (151, 216, and 367 base pairs) or RR (367 base pairs).

Line 137/Line 162: it is hard to understand how the authors did statistical analyses:

-          Could the authors explore how Tukey-Kramer method was used

The average value of the measurement at 1-3 dpi for each individual was first calculated. After performing a Bartlett test and analysis of variance based on the average value for each individual, the Tukey-Kramer method was used to make multiple comparisons for differences in all combinations.

-          Why the authors select Fisher’s exact test in this case, which is the test group (which one versus which one).

We utilized the Fisher’s exact test since the expectation was less than 5 in the 2 × 2 contingency table test. We followed the statistical methods of the following previously published report: PMID: 31528339.

-          Why did the authors use the t-test and which type of t-test?

In the original manuscript, the measurement of the duration of diarrhea was performed with the exclusion of dead pigs; therefore, the multiple comparisons test could not be run in the test groups due to high mortality. Thus, we chose to use the student's t-test. In the revised manuscript, however, we avoided statistical analysis for the duration of diarrhea due to the lack of appropriate analysis.

-          Did the authors consider the interaction between the strain and genotypes?

Yes, we considered interactions between strains, amount of ETEC used for infection, and genotypes, and we analyzed each combination of factors.

Table 3; The authors indicate the rank, not the mean in the second column

We have modified the text in the revised manuscript according to the Reviewer’s recommendation, replacing the mean with range in the second column of Table 3.

Line 161/Line 198-200: I did not convince the author's possibility to use one SNP to define the animals as Susceptible or resistant to disease. Moreover, the authors should provide more details about this marker in the introduction and method sections

We would like to clarify that we never made the conclusion that this SNP in the MUC4 gene controls the development of PWD. Our intention was to highlight the evidence that MUC4 polymorphisms have a significant impact on PWD. In our study in particular, we have demonstrated the importance of MUC4 genotype by showing that the administration of 1010 CFU/pig of highly adherent E. coli results in diarrhea in 100% of RS pigs regardless of breed.

Table 4” Please bring the SE

We added SE values in Table 4, as recommended by the Reviewer.

Did the authors have the PCR-RLFP picture? *Since the authors have very hard to adjust the changes in the experiments are true or not

Please see the attachment for an image of PCR-RLFP agarose gel. We did not think that it was necessary to include this image in the manuscript, but we can do so if the Reviewer sees fit.

Table 4: it is not clear differences between the

We are not sure what the Reviewer is referring to, but we would be happy to address any additional concerns they may have.

Table 5: How many pigs in each breed

Please add the footnote for the breed name

We have modified the text in the revised manuscript according to the Reviewer’s recommendation, adding the number of animals used in Table 5.

Line 206/Line 223: What is the mean of Bars and SE

We have corrected the text in the revised manuscript, replacing it with “Error bars, SE.”

Line 213/Line 255: replace “studies by Matsuda et al. and Jin et a report” Matsuda et al. and Jin et al previously reported

We have modified the text in the revised manuscript according to the Reviewer’s recommendation

Line 214: Why the stx2e gene can cause the difference

  1. coli carrying the stx2e gene also often carries the F18 fimbriae gene. This study was performed using E. coli that do not carry the stx2e gene; therefore, we believe that E. coli with the F4 fimbriae gene may be the predominant species involved in PWD compared to those with the F18 fimbriae gene.

Line 218 219/Line 260-262: please re-write

It is not known how much each individual enterotoxin contributes to PWD, but it is believed that each enterotoxin can cause diarrhea [18–20].

For clarification, we have modified the revised manuscript as follows:

It is still unclear the contribution of each individual enterotoxin strain to PWD; however, it is believed that all strains can cause diarrhea [18–20]. Therefore, we examined cell adhesion ability in strains carrying all toxin genes.

Line 230-231/Line 273-274: The authors did not need to repeat the results

As recommended by the Reviewer, we have modified the text in the revised manuscript as follows:

In SS type pigs infected with 1010 CFU/pig of highly adherent ETEC, severe diarrhea (watery stool) was observed at 1-3 dpi but not at 4-7 dpi (Table 4).

Table 5: There is some difference between the WL and LL in the mean fecal score, duration, and mortality? What did the authors could explain it?

Significant differences were observed between WL and LL in mean fecal scores at 1-3 dpi and 4-7 dpi. In contrast, no differences were found in terms of duration or mortality. The cause of the difference in the mean fecal scores between WL and LL is unclear. One possibility is that polymorphisms other than this particular MUC4 SNP could be involved in the etiology of PWD. That said, at 1-3 dpi, our PWD model resulted in diarrhea in all RS pigs regardless of pig breed; based on these results, we believe that this model could be very useful for the development and testing of PWD countermeasures.

Round 2

Reviewer 1 Report

No further comments.

Author Response

We are very grateful to you for peer review.

Reviewer 2 Report

The revised manuscript is improved. I still have a major concern about the experimental design as I did not agree about the selection and use of MUC4 as a candidate biomarker to test the sensitivity of PWD.

There are still issue with the writing and formatting in general. The authors did not pay enough attention to this point. 

Line 22: The author used only one polymorphism in the gene, it might be good if the author mentions this polymorphism 

Line 22: Please do not use abbreviation at the beginning of a sentence

LIne 74 and others: Add the full company name, city and country for each equipment

Line 145: Some abbreviations such as DPI should be defined at the first appeared in the manuscript. 

Author Response

The revised manuscript is improved. I still have a major concern about the experimental design as I did not agree about the selection and use of MUC4 as a candidate biomarker to test the sensitivity of PWD.

Response:

We appreciate the Reviewer’s feedback. As mentioned in the response to the reviewer’s comments on the original manuscript, the purpose of this study was to develop a successful PWD infection model, not to investigate which genes are predominantly involved in the onset of PWD. The reason why we chose MUC4 gene was described in the text (underline shows a newly added description):

The gene mucin 4 (MUC4) encodes for the F4 fimbriae receptor. The MUC4 gene has been identified on chromosome 13 [6]. Recently, a DNA marker-based test targeting MUC4 has become available to genotype pigs and identified single nucleotide polymorphism (SNP) (DQ848681:g.8227C>G) associated with F4ab/ac ETEC resistance and susceptibility [7]. Using this DNA marker-based test to genotype for MUC4, pigs can be categorized into the following three groups in terms of their susceptibility to ETEC-induced diarrhea: resistant (RR), susceptible heterozygote (SR) and susceptible homozygote (SS). While the incidence of diarrhea was increased by ETEC infection, sensitivity to PWD was dependent on MUC4 genotype and the occurrence of diarrhea did not reach 100% [8,9]. As differences in MUC4 genotypes appear to affect the composition of gastrointestinal microbiota, MUC4 polymorphisms have been proposed to play roles in intestinal infection [10], such as PWD.

Indeed, we succeeded in achieving 100% incidence of diarrhea in RS-type pigs regardless of breed by administering highly adherent E. coli at 10(10) CFU/pig. Using a PWD infection model with high accuracy for the development of diarrhea, such as our method, we are better equipped at developing countermeasures against PWD.

There are still issue with the writing and formatting in general. The authors did not pay enough attention to this point.

The original and revised manuscripts were reviewed by a native English speaker/writer with a Ph.D., and the second, revised text was also reviewed by the same individual.

Line 22: The author used only one polymorphism in the gene, it might be good if the author mentions this polymorphism

As mentioned above, we added information regarding SNP in the text (Introduction).

Line 22: Please do not use abbreviation at the beginning of a sentence

We have modified some sentences in the text in the revised manuscript according to the Reviewer’s recommendation.

LIne 74 and others: Add the full company name, city and country for each equipment

We have modified the text in the revised manuscript according to the Reviewer’s recommendation.

Line 145: Some abbreviations such as DPI should be defined at the first appeared in the manuscript. 

While DPI was defined at the first appearance in the text (Line 145), we found other abbreviations without definition. We defined all abbreviations.
